# Probabilistic Analysis of Critical Speed Values of a Rotating Machine as a Function of the Change of Dynamic Parameters

**DOI:** 10.3390/s24134349

**Published:** 2024-07-04

**Authors:** Zdenko Šavrnoch, Milan Sapieta, Vladimír Dekýš, Petr Ferfecki, Jaroslav Zapoměl, Alžbeta Sapietová, Michal Molčan, Martin Fusek

**Affiliations:** 1Department of Applied Mechanics, Faculty of Mechanical Engineering, University of Zilina, Univerzitná 8215/1, 010 26 Zilina, Slovakia; zdenko.savrnoch@fstroj.uniza.sk (Z.Š.); vladimir.dekys@fstroj.uniza.sk (V.D.); alzbeta.sapietova@fstroj.uniza.sk (A.S.); 2Department of Applied Mechanics, VŠB-Technical University of Ostrava, 17. listopadu 15/2172, 708 00 Ostrava, Czech Republic; petr.ferfecki@vsb.cz (P.F.); jaroslav.zapomel@vsb.cz (J.Z.); michal.molcan@vsb.cz (M.M.); martin.fusek@vsb.cz (M.F.); 3IT4Innovations, VŠB-Technical University of Ostrava, 17. listopadu 2172/15, Poruba, 708 00 Ostrava, Czech Republic

**Keywords:** rotordynamics, Monte Carlo method, vibration analysis, nonlinear dynamics, uncertainty analysis, Jeffcott rotor, Laval rotor

## Abstract

Real-world rotordynamic systems exhibit inherent uncertainties in manufacturing tolerances, material properties, and operating conditions. This study presents a Monte Carlo simulation approach using MSC Adams View and Adams Insight to investigate the impact of these uncertainties on the performance of a Laval/Jeffcott rotor model. Key uncertainties in bearing damping, bearing clearance, and mass imbalance were modeled with probabilistic distributions. The Monte Carlo analysis revealed the probabilistic nature of critical speeds, vibration amplitudes, and overall system stability. The findings highlight the importance of probabilistic methods in robust rotordynamic design and provide insights for establishing manufacturing tolerances and operational limits.

## 1. Introduction

In their scientific article, Fu et al. [1] propose an innovative approach that enables the analysis of vibrations experienced by rotors under conditions of uncertainty. The proposed method utilizes the Legendre series to model the uncertain parameters of the rotor system, thereby allowing for a more accurate representation of the system’s dynamics. The method is applied to a case study of a Jeffcott rotor, and the results demonstrate its effectiveness in capturing the effects of uncertainty on the rotor’s vibration response. The proposed method provides a valuable tool for the vibration analysis of rotors in practical applications, where uncertainties are often encountered.

Chapter 24 of the article offers a comprehensive overview of nonlinear reduced order model (ROM) techniques for nominally cyclic symmetric structures and rotating machinery. It introduces fundamental concepts of nonlinear ROM and focuses on two specific methods: the Harmonic Balance Method (HBM) and the Proper Orthogonal Decomposition (POD) method. The chapter concludes with discussions on challenges and future directions of nonlinear ROM, including the need for efficient and accurate methods, handling complex nonlinearities, and model validation. Future directions involve developing new methods, applying ROM to new problems, and utilizing it in design and optimization [2].

In conclusion, the study investigates the influence of support parameters on the vibrations of a cracked rotor passing through its critical speed. The results indicate that the support parameters have a substantial impact on the rotor’s vibrations, particularly the spring stiffness and damping coefficient. The natural frequencies and damping ratios of the rotor are heavily influenced by these parameters. The support spacing has a lesser impact on vibrations but can influence rotor stability. These findings are essential for designing support systems for cracked rotors to reduce vibrations and guarantee safe operation. Future research should focus on extending the model to include other factors, such as the crack depth and location, as well as investigating the effects of nonlinear support characteristics [3].

Authors Briend Y. and colleagues [4], in their scientific article, explore the complex dynamics of a rotor system under varying support conditions. They use a combination of theoretical analysis and experimental measurements to investigate the rotor’s behavior as it passes through a critical speed, where the system’s natural frequency coincides with the excitation frequency. The study focuses on a rotor supported by a combination of hydrodynamic bearings and ball bearings with loose fit. This configuration introduces nonlinearities and clearances into the system, making its dynamic behavior more complex. The authors employ a multi-degree-of-freedom model to capture the nonlinear effects and simulate the rotor’s response. Analysis reveals that the presence of loose-fit ball bearings significantly influences the rotor’s dynamics. The impact of the loose fit is particularly pronounced at the critical speed, where it can lead to increased vibration amplitudes, subharmonic resonances, and chaotic behavior. The study also highlights the role of bearing clearance and damping in shaping the rotor’s response. Findings of this research have implications for the design and operation of rotor systems in various industries, such as power generation, transportation, and manufacturing. Understanding the nonlinear dynamics of rotors with loose-fit bearings is crucial for predicting their behavior, avoiding potential instabilities, and ensuring safe and reliable operation [4].

In their study, Han Y. et al. [5] investigate the effects of nonlinearities on the dynamic behavior of rotor systems with residual shaft bow. The authors use a combination of theoretical analysis and numerical simulations to study the influence of various nonlinear factors, such as bearing clearance and foundation flexibility. The results show that nonlinear factors can significantly affect the response characteristics of rotor systems, including the critical speeds, amplitudes, and stability of the system. In particular, the presence of a residual shaft bow can lead to more complex and unpredictable behavior, making it more difficult to control and maintain the stability of the system. The findings of this study have important implications for the design and operation of rotor systems in various industrial applications.

Fu C. et al. [6] delve into the significance of unbalance detection in rotating machinery and the subsequent consequences of an imbalance. It introduces polar and orbit plots as useful tools for visualizing and analyzing the behavior of a rotating system. The authors present a detailed description of polar and orbit plots, including their construction and interpretation. They emphasize the importance of phase angles in understanding the rotational motion of the system. The paper also discusses the application of polar and orbit plots in unbalance identification. It explains how the plots can be used to determine the magnitude and angular position of the unbalance and provides practical examples to illustrate the analysis process. Overall, the article offers a comprehensive overview of polar and orbit plot analysis, highlighting their value in the diagnosis of unbalance in rotating systems.

Pelaez G. and colleagues [7], in their research paper, propose a novel approach for estimating parameters in rotating machinery systems with uncertainties. The proposed method is based on a combination of algebraic and model-based techniques, which allows for the identification of parameters even in the presence of significant uncertainties. The method is applied to a case study of a rotor-bearing system, and the results demonstrate its effectiveness in estimating parameters with high accuracy and robustness. The proposed method provides a valuable tool for the condition monitoring and fault diagnosis of rotating machinery systems, where uncertainties are often encountered.

The scientific article presented by Gasch R. [8] presents a comprehensive investigation of the dynamic behavior of a Laval rotor with a transverse crack. Using a combination of theoretical analysis and experimental measurements, the study aims to understand the effects of the crack on the rotor’s vibration response and stability. Gasch employs a finite element model to simulate the cracked rotor system and analyzes the influence of crack parameters such as depth, location, and orientation on the rotor’s natural frequencies, damping ratios, and mode shapes. The study also examines the effects of operating conditions, including rotational speed and unbalance, on the rotor’s dynamic behavior. The findings of the study provide valuable insights into the dynamic characteristics of cracked rotors and their potential for failure. The results indicate that the crack can significantly alter the rotor’s natural frequencies and mode shapes, leading to increased vibration levels and reduced stability. The study emphasizes the importance of early crack detection and monitoring to prevent catastrophic failures in rotating machinery [8].

Prohl M., in his paper [9], marks the development of the transfer matrix method. The paper presents a comprehensive method for determining the critical speeds of flexible rotors. The method is based on the finite element method and takes into account the effects of flexibility, damping, and gyroscopic moments. The method is applied to a number of examples, including a Jeffcott rotor and a Laval rotor, and the results are compared with experimental data. The method is able to accurately predict the critical speeds of flexible rotors, even in cases where the rotors are highly flexible and have complex geometries. The method is also able to predict the effects of damping and gyroscopic moments on critical speeds. The presented method is a valuable tool for the design and analysis of flexible rotors. It can be used to determine the critical speeds of rotors, to identify the sources of vibration, and to develop strategies for reducing vibration.

Saucedo-Dorantes et al. (2021) present a novel condition-monitoring methodology for induction motors, leveraging a high-dimensional set of hybrid features extracted from vibration and stator current analyses. The proposed approach utilizes artificial intelligence and machine learning techniques to optimize and reduce the feature space, enhancing the identification of multiple and combined faults occurring simultaneously. The effectiveness of the methodology is validated through a comprehensive experimental dataset encompassing healthy, single-fault, and combined-fault conditions [10].

Unbalance in rotating machinery is a prevalent issue in industrial settings. It leads to excessive vibrations that create unwanted noise, accelerate wear on components, and shorten the machine’s lifespan. These vibrations can propagate through the system, affecting other connected equipment and potentially causing catastrophic failures [11,12]. The design and operation of rotating machinery critically depend on accurate prediction and mitigation of vibration, especially at critical speeds where resonance can result in catastrophic failure. However, traditional deterministic approaches often cannot capture the inherent uncertainties associated with real-world rotor systems, such as variations in material properties, manufacturing tolerances, and operating conditions. These uncertainties can significantly affect the dynamic response and stability of the rotor. Therefore, it is essential to develop robust methods to quantify uncertainties.

The machine learning technique offers a more general approach for nonlinear systems, potentially applicable to a wider range of machinery but requiring a substantial amount of training data, prolonging the analysis process [13]. The choice between these methods would depend on the specific application and the available resources. If a comprehensive analysis of multiple and combined faults is required, the approach by Saucedo-Dorantes et al. (2021) might be more suitable [10].

However, if computational resources are limited or the focus is on vibration-based fault detection, the method presented in this article/paper could be a viable alternative. The proposed technique can be easily adapted to different types of uncertainties, including stochastic uncertainties and uncertainties caused by a lack of knowledge of different parts parameters. This flexibility makes it a versatile tool for analyzing a wide range of rotor systems and operating conditions. By modeling key uncertainties using probability distributions, this technique provides a realistic representation of real rotordynamic systems that are naturally affected by changes and uncertainties. Data obtained from the analysis can be used to determine manufacturing tolerances and operating limits, leading to more robust and reliable rotordynamic designs.

The novelty and contribution are considered to be the probabilistic interpretation of the system output parameters, such as the critical frequency, which takes into account the dispersion in the values of the input parameters of the computational system or the uncertainty with which the parameters of the real system are determined by the measurement. This probabilistic approach allows for the estimation of, for example, the risk that the critical speed will exceed a certain specified value, and so on. We consider the approach described above to be applicable, for example, also in the case of analysis of the consequences of manufacturing tolerances for newly manufactured objects or for objects in use that have changed their default parameters, e.g., as a result of degradation processes.

### 1.1. Disk

The disk is considered to be rigid when its position remains normal relative to the shaft during the vibration. Essentially, the motion of a rigid disk can be described by the motion of the rotor station (shaft) to which it is attached. The disk itself does not introduce any new DOFs to the system.

Flexible disk, on the other hand, can tilt and its rotational displacements can differ from the rotor station to which it is attached to.
(1)$$q={\left(x,y,\theta_{x},\theta_{y}\right)}^{T}$$

A flexible disk can greatly impact overhung rotors with large inertia by reducing the gyroscopic stiffening effect. Considering this, flexibility is essential in applications like large gas turbines and fans, where it significantly influences rotor behavior.

### 1.2. Bearings

Bearings play a crucial role in shaping the dynamic behavior of a rotor system. The reality is that all bearing forces exhibit nonlinear characteristics, adding a layer of complexity to the system’s response [14].

In rotordynamics, ‘bearings’ are any components connecting rotating and non-rotating structures. This includes traditional bearings (fluid-film, rolling element, magnetic), as well as dampers, seals, supports, and even fluid-induced forces. The key characteristic of a bearing in this context is that it connects two points without adding new degrees of freedom to the system.

### 1.3. Excitations

Mass unbalance is the primary source of synchronous excitation in rotating machinery. This unbalance creates a harmonic excitation at the rotor’s rotational frequency (1 × Ω). Importantly, the magnitude of this excitation increases with the square of the rotor speed (meΩ^2^), where m is the mass, e is the eccentricity, and Ω is the rotational speed.

Other sources of synchronous excitation also exist, including disk skew, shaft bow, misaligned couplings, and magnetic forces. It is interesting to note that the excitation moments caused by disk skew (I_p_ − I_d_) τ Ω^2^ share similarities with mass unbalance forces (Figure 1). Both increase with the square of rotor speed and excite the system at the rotational frequency.

Disk skew produces an external moment (gyroscopic moment), which can significantly affect rotor stability, especially at high speeds. Mass eccentricity produces an external force, and this force acts directly on rotor.

There are also other types of excitations and loadings that exist in rotor-bearing systems. These include constant forces (such as static and gravitational loads), frequency- or speed-dependent excitations, and transient excitations exhibiting time-dependent behavior.

## 2. System Equations of Motion

The governing equations of motion for a complete rotor-bearing-support system are obtained by assembling the equations of motion of all the components, often derived using the Lagrangian approach.

The equation of motion is determined following:(2)$$\frac{d}{dt}\left(\frac{\partial L}{\partial \dot{q}}\right)-\frac{\partial L}{\partial q}+\frac{\partial R}{\partial \dot{q}}=0$$
where L is the Lagrangian and it is defined as the difference between the kinetic energy T and potential energy U.
(3)L=T−U

R is the dissipation function and represents the energy dissipation in the system due to internal friction and damping.

q is the generalized displacement vector.

The kinetic and potential energy terms can be readily determined using either the Timoshenko or the Bernoulli–Euler beam theories. The level of the kinetic energy is determined by both the lateral and the rotatory inertial effects of the shaft and disk. Lagrange equation is expanded using the energy formula to derive a vector differential equation characterizing motion of the system.
(4)$$M\ddot{q}+C\dot{q}+G\dot{q}+Kq=F_{ex}$$
where: 

M is the mass matrixG is the gyroscopic matrixK is the stiffness matrixC is the damping matrix$F_{ex}$ is the generalized external force (or Torque)

Gyroscopic matrix is defined as
(5)$$G=\left[\begin{array}{cc} \begin{array}{cc} 0 & 0\\ 0 & 0 \end{array} & \begin{array}{cc} 0 & 0\\ 0 & 0 \end{array}\\ \begin{array}{cc} 0 & 0\\ 0 & 0 \end{array} & \begin{array}{cc} 0 & \Omega I_{p}\\ -\Omega I_{p} & 0 \end{array} \end{array}\right]$$
where *I_p_* is the polar moment of inertia of the uniform rigid cylindrical rotor and Ω is rotational speed.

The kinetic energy comes from the translational and angular momentum of the rotor system.
(6)$$T_{i}=\frac{1}{2}{\dot{q}}_{i}^{T}M_{i}{\dot{q}}_{i}+\frac{1}{2}{\omega \dot{q}}_{i}^{T}W_{i}{\dot{q}}_{i}$$
(7)$$T=\frac{1}{2}m_{d}\left({\dot{x}}^{2}+{\dot{y}}^{2}\right)+\frac{1}{2}I_{d}\left({\dot{\theta }}_{x}^{2}+{\dot{\theta }}_{y}^{2}\right)+\frac{1}{2}\Omega I_{p}\left({\dot{\theta }}_{x}\theta_{y}-\theta_{x}{\dot{\theta }}_{y}\right)$$
where W is the moment of inertia matrix with rotational speed.

The potential energy of the system is based on the Bernoulli–Euler beam theory and comes from the internal strain energy of the shaft due its lateral bending.
(8)$$U_{i}=\frac{1}{2}q_{i}^{T}K_{i}q_{i}$$
where K is the stiffness matrix which describes axial stress/strain due to the lateral shaft bending.

The energy dissipation in the shaft due to internal friction is usually small and can be neglected. However, in certain cases where the dissipation function cannot be ignored, the expression for $R_{i}$ assumes a specific form.
(9)$$R_{i}=\frac{1}{2}{\dot{q}}_{i}^{T}C_{i}{\dot{q}}_{i}$$
where: C is the damping matrix.

Combining these terms in Lagrange equation we obtain the vector differential equation.
(10)$$M_{i}{\ddot{q}}_{i}+C_{i}{\dot{q}}_{i}+G_{i}{\dot{q}}_{i}+K_{i}q_{i}=F_{i}$$

The system governing equations of motion is derived by combining the individual equations of motion for each component. When the rotational speed (Ω) is constant, the resulting equation can be expressed as follows:(11)$$M\ddot{q}\left(t\right)+\left(C+\Omega G\right)\dot{q}\left(t\right)+Kq\left(t\right)=Q\left(t\right)+Q_{nb}\left(\dot{q,q,t}\right)$$

In which all matrices are real and assembled from the associated component. Where q is the system displacement vector or generalized coordinate vector needed to be solved.

Sometimes in rotating machinery, we need to analyze what happens when the rotor is not spinning at a constant speed, like during startup or in certain emergency situations. This requires different equations to handle the changing rotational speed and how it influences the forces and vibrations in the system. This leads to the governing equations of motion.
(12)$$M\ddot{q}\left(t\right)+\left[C+\dot{\varphi }G\right]\dot{q}\left(t\right)+\left[K+\ddot{\varphi }G\right]q\left(t\right)\;\;\;\;\;\;\;\;\;\;\;\;\;\;\;\;={\dot{\varphi }}^{2}Q_{1}\left(\varphi \right)+\ddot{\varphi }Q_{2}{\left(\varphi \right)+Q}_{nl}\left(\dot{q},q,\varphi ,\dot{\varphi },\ddot{\varphi },t\right)$$
where: 

*M* is the mass matrix*C* is the damping matrix*G* is the gyroscopic matrix*K* is the stiffness matrix$Q_{nl}$ is the force vector$\varphi ,\dot{\;\varphi },\ddot{\;\varphi }$ are angular displacement, angular velocity and acceleration

Initial conditions:(13)$$q\left(0\right)=q_{0},\;\;\;\;\dot{q}\left(0\right)={\dot{q}}_{0}$$

Due to the changing speed, additional components in Equation (12) appear the circulatory matrix ($\ddot{\varphi }G$) and the forcing vector ($\ddot{\varphi }Q_{2}$). This makes the equations nonlinear, so we need time-transient analysis to solve them.

### 2.1. Adams Insight

Adams Insight is an extension of the Adams (2023.4) product family. It is a design-of-experiments (DOE) software, which allows the design of sophisticated experiments for measuring the performance of mechanical systems. It also contains a portfolio of statistical tools for analyzing the results. Experimental design, often referred to as Design of Experiment (DOE), is a compilation of statistical tools and procedures used to plan experiments and analyze their outcomes effectively. These experiments are typically conducted to evaluate the performance of prototypes, manufacturing processes, and the quality of finished products. Initially, they were developed for physical experiments but are also equally effective when applied to virtual experiments.

### 2.2. Monte Carlo Method

This method randomly sets values of the specified design factors for each run of the simulation. The investigation aims to evaluate the impact of real-world variations on the design’s performance [15,16,17]. By conducting numerous trials, statistical predictions regarding the design’s response can be established. The method’s foundation lies in representing parameters using a Probability Density Function (PDF). For each parameter subjected to variation in the analysis, a specified PDF is required.

Investigation Strategy chosen to be Monte Carlo method and Design Type—Full Factorial. It is a comprehensive method. It incorporates every potential combination of levels (number of possible values that can be taken by a design factor) for each design factor. The number of runs required follows a mathematical formula: m^n, where “m” is the number of levels and “n” represents the number of factors. However, it is crucial to note that as the values of “m” and “n” escalate, Full Factorial becomes more appropriate for experiments involving a limited number of factors as the number of runs increases exponentially. Based on the specified design type, Adams Insight produces a design matrix.

## 3. Model Setup

This study utilizes a Laval/Jeffcott rotor model to investigate solver performance in MSC.ADAMS View (Figure 2). Key model components and their characteristics are as follows:Shaft: The shaft (ϕ15-L550) is modeled as a uniform rod with a circular cross-section. It is represented as a flexible body using the View Flex function in Adams (2023.4) software. Material of the shaft is steel, which is specified by the Adams material library. The Young’s modulus of the steel is 207 (GPa), the Poisson’s ratio is 0.29, and the density is 7801 (kg/m^3^);Rigid Disk: A mass imbalance is introduced by a single rigid disk mounted on the flexible shaft at a distance equal to 1/4 of the bearing distance. The disk’s mass is 1.41 kg, and its moments of inertia around the x, y, and z axes are I_xx_ = 1980.0672 (kg·mm^2^), I_yy_ = 1123.5965 (kg·mm^2^) and I_zz_ = 1123.579 (kg·mm^2^);Bearings: Shaft is supported by two SKF 61805 deep groove ball bearings with C0 clearance. Bearings are modeled with Adams Bearings Plugin (Machinery toolbox) as detailed representations, incorporating their stiffness, potential nonlinearities, and realistic contact behavior.Shaft shackles: Aluminum elements between bearings and shaft acting like a shaft adapter.

Adams Dynamic Solver was preset to Newmark integrator. GSTIFF was considered as well but while implicit solvers are generally better at handling nonlinearities than explicit ones, the Newmark method is still a step above pure implicit solvers, often allowing simulations to converge and find solutions where a solver like GSTIFF might struggle.

### 3.1. Methodology

In order to thoroughly assess the behavior of the four distinct rotor configurations, Monte Carlo simulations were implemented. During these simulations, important input parameters were systematically varied across a substantial number of trials [18]. This approach yielded datasets that provided a basis for subsequent analysis (Figure 3).

To gain insights into the complex interdependencies between various parameters, scatterplot matrices were pictured. These visual representations effectively depicted the relationships between different parameters, enabling the identification of patterns and potential correlations. Additionally, histograms were utilized to illustrate the distribution of critical frequencies, providing valuable insights into the range and dispersion of these frequencies across the different configurations.

The combination of scatterplot matrices and histograms facilitated a comprehensive understanding of the behavior and performance of the four-rotor configurations. Leveraging these visualizations, subtle nuances and variations that might have otherwise remained undetected can be found, contributing to a better understanding of the investigated system.

### 3.2. Bearing Damping and Clearance

According to our simulations, the bearing damping and clearance had a minimal impact on the critical frequency for the balanced system with disk in central position, respectively. This challenges conventional assumptions about their roles in rotor dynamics [19,20,21]. Damping and clearance might be less influential than previously thought (other parameters like unbalanced mass is more influential). Further research is needed to understand their effects fully.

### 3.3. Position of an Unbalanced Mass

The unbalanced mass on the experimental model was added and positioned, as is shown in Figure 4.

This experimental model was used for the validation of numerical models (Figure 5).

The analysis of shaft orbitals was performed on the experimental model (Figure 6). This analysis was preferred over the analysis of critical speed due to the excessive vibrations and shaft deflections threatening the damage of sensors, other equipment, and the experimental model itself. Analysis was performed for the several rotational speed regimes with 6.1 [g] of unbalanced mass. Comparison of trial run with rotational speed of 377 [rad/s] (60 [Hz]) with numerical model (Adams) is provided (Figure 7).

Both trajectories have elliptical shapes. Despite the minor differences, the overall agreement between the simulated and measured orbits is quite good, however there is still room for the numerical model improvement [22]. This shows that the simulation model captures the essential characteristics of the motion reasonably well.

Due to the fact that there is never a 100% precision rate in manufacturing and assembly processes there is always an amount of uncertainty present in the system influencing its performance [23].

Parameters that were considered to be the most influential on the rotor system behavior were chosen as the input parameters for this analysis [24,25]. The influence of the ambient temperature was not considered in the analysis. In order to thoroughly assess the behavior of the distinct rotor configurations, Monte Carlo simulations were implemented. Matrix of input values (Table 1) for the Monte Carlo analysis was generated with variables (input arguments) as follows:

The same matrix of input arguments was used for different rotor configurations. Configurations with additional unbalanced mass were considered and evaluated to determine the influence of these input arguments on the rotor performance, namely on the rotor critical speed. Shaft deflection in three geometrical axes (x—axial, y—vertical, z—horizontal) and deflection vector magnitude. The width of the intervals may be influenced by the scales and measurement methods used, which we do not consider in this paper. For example, the interval for bearing clearance was taken from the C0 tolerance field of SKF bearings.

To gain insights into the complex interdependencies between various input parameters, scatter plot diagrams were generated (Figure 8 and Figure 9). These visual representations effectively depicted the relationships between different parameters, enabling the identification of patterns and potential correlations [26].

The relationship between shaft deflection and critical velocity has an approximately “linear” character, with a greater dispersion of values being evident.

A positive correlation between unbalanced mass (m) and shaft magnitude deflection (Sm) is expected. This is because a larger unbalanced mass would cause larger centrifugal forces, leading to greater deflection of the shaft [27].

The relationship between unbalanced mass and the individual deflection components in the x, y, and z axes (Sx, Sy, Sz) is evident (with a wider spread on the y-axis). It depends on the specific location and orientation of the unbalanced mass relative to the shaft’s principal axes.

For the rotor 025un configuration (Figure 9), the dependency can also be approximated as a “linear” function; however, compared to the rotor 05un configuration (Figure 8), a greater influence of the stabilization effect is present, as a result of the gyroscopic effect.

Additionally, histograms were used to illustrate the distribution of critical frequencies, providing valuable insights into the range and dispersion of these frequencies across the different configurations.

The histograms comparing critical speed distributions for two-rotor configurations (05un and 025un) reveal the impact of unbalanced mass (Figure 10). Rotor 05un configuration, with the mass at the center of the shaft, exhibits a narrower range of critical speeds centered around 811 [rad/s], indicating higher predictability and less sensitivity to mass variations. In contrast, Rotor 025un, with a mass at 0.25 of the shaft length, displays a wider distribution of critical speeds, ranging from 1100 to 1200 [rad/s], suggesting greater variability and a stronger mass dependence. These findings depict the importance of unbalanced mass location in rotor system design. The results suggest that the presence of an unbalanced mass in the relative center of the rotor shaft leads to a more predictable and stable system but with lower critical speed. Although placing it off-center increases operational flexibility, careful consideration of mass distribution is required to mitigate potential resonance risks [28].

## 4. Results

A strong positive correlation was found between mass (m) and critical speed (Ω_crit_) of unbalanced systems (025-un, 05-un), indicating that heavier rotors generally exhibit higher critical speeds [9]. On the other hand, in the balanced systems (025-bal, 05-bal), other factors play more dominant roles in determining the critical speed.

The critical speed increases with unbalanced mass, which is visible in the upward trend in this diagram (Figure 11). There is a clear positive correlation between mass and critical speed. This indicates a good degree of predictability for this rotor configuration.

This graph depicts the influence of the unbalanced mass on the rotor critical speed and the probability of critical speed occurrence (Figure 12).

In the 025-rotor configuration, the critical speed is directly proportional to the unbalanced mass. This means that as the unbalanced mass increases, so does the critical speed. This behavior is consistent with the understanding that the larger the unbalanced mass, the greater the centrifugal force acting on the rotor, which in turn leads to higher vibration amplitudes and a higher critical speed. Compared to the 05-rotor configuration, the 025-rotor configuration has a larger critical speed range.

In conclusion, these two sets of diagrams estimate the density probability function for unbalanced mass (m) and critical speed (Ω_crit_) as an approximately “linear” function, with mass having a greater impact on the critical speed of rotor 025un configuration.

Boxplot (Figure 13) provides a summary of the critical speed distributions for the analyzed rotor configurations. It highlights the significant influence of unbalanced mass on the critical speed and its variability.

The critical speed for rotor05un is distributed around 810 rad/s (approximately 129 Hz). The interquartile range (IQR), which represents the spread of the middle 50% of the data, is relatively small, indicating that the critical speeds for this configuration are fairly consistent (Table 2).

The critical speed for rotor025un is distributed around a much higher value, approximately 1145 rad/s (182.3 Hz). The IQR is larger than that of rotor05un, suggesting higher influence of unbalanced mass on the critical speed of this rotor configuration. The rotor025un configuration also shows greater variability in critical speeds compared to rotor05un. This suggests that the location of the unbalanced mass not only affects the average critical speed but also the range of possible critical speeds (Table 2).

## 5. Conclusions

This study provides a procedure for estimating the critical velocity of a flexible rotor with a rigid disk considering uncertain input parameters expressed in terms of a probability density function on a selected interval (normal distribution was used, and input data were generated in Adams Insight). The rotor run-up defined by the Step function in the Adams environment was simulated. The rotor deflection was registered, and the critical speed was determined by postprocessing. The critical rate estimate was in the form of a 95% confidence interval. This procedure allows us to predict the value of the critical velocity. A similar procedure can be applied to the deflection of the shaft during the run-up, which can be used, for example, when adjusting the position of the vibration sensor due to the recommended distance from the rotor and the risk of contact between the rotor and the sensor.

This study provides valuable insights into the relationship between various parameters and the critical frequency in balanced and unbalanced rotor systems. The findings emphasize the importance of considering mass distribution and potential non-linear effects for accurate prediction and optimization of rotor system designs. The combination of scatterplot matrices and histograms facilitated a comprehensive understanding of the behavior and performance of the four-rotor configurations. Leveraging these visualizations, subtle nuances, and variations, which might have otherwise remained undetected, contributed to a better understanding of the investigated system. 

Although the proposed simulation approach for analyzing uncertainty propagation in Laval and Jeffcott rotors offers several advantages, it also has limitations. The proposed approach still requires significant computational resources, especially for complex rotor systems with many parameters affected by uncertainty. The number of simulations needed for accurate results is large, leading to increased computational time and cost. Also, the approach quantifies uncertainties in rotor system input parameters like material properties, dimensions, and imbalance but may not address other sources like model uncertainties or external factors. The accuracy of the simulation depends on the accuracy of the model assumptions. Simplification of rotor geometry, materials, and boundary conditions can cause inaccuracies. For reliable simulations, it is essential to consider these assumptions.

Despite these limitations, the proposed simulation approach provides another tool for understanding and mitigating the impact of uncertainties in rotor systems. Further research, incorporating non-linear models and experimental validation, is warranted to comprehensively understand and mitigate the risks associated with rotor system dynamics.

## Figures and Tables

**Figure 1 sensors-24-04349-f001:**
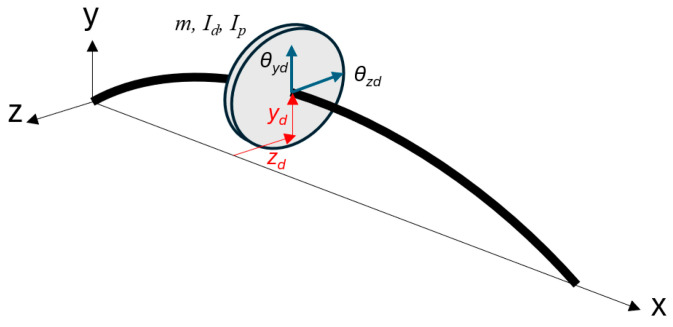
Schematical representation of an analyzed model.

**Figure 2 sensors-24-04349-f002:**
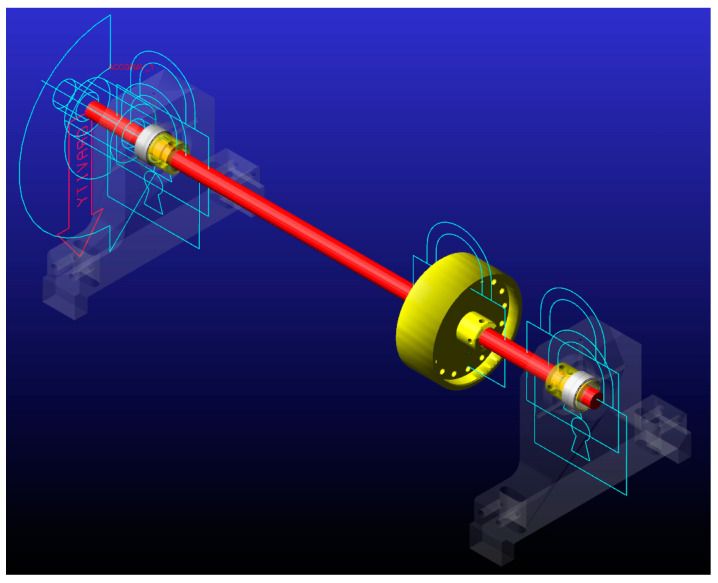
Numerical model analyzed in Adams View (025un configuration).

**Figure 3 sensors-24-04349-f003:**
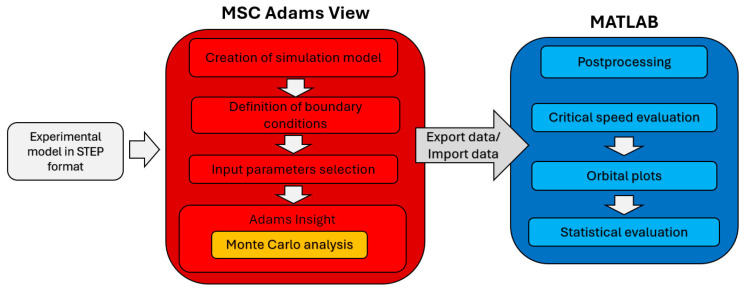
Graphical representation of the workflow.

**Figure 4 sensors-24-04349-f004:**
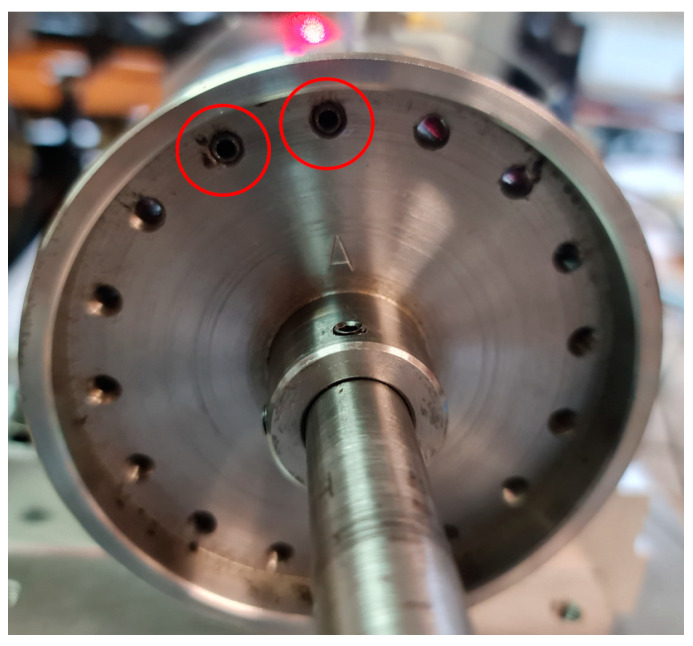
Location of unbalanced mass.

**Figure 5 sensors-24-04349-f005:**
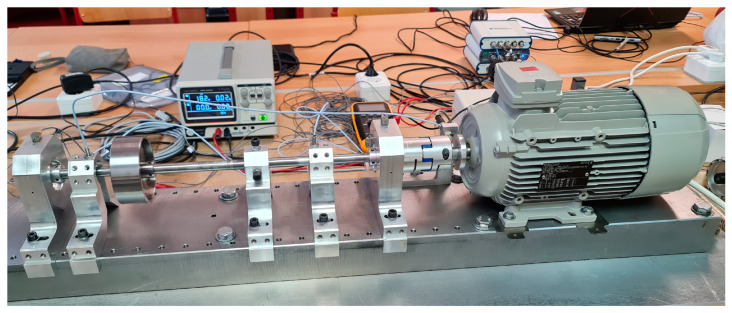
Rotor with the 025un configuration (Disk mounted in the center of the shaft).

**Figure 6 sensors-24-04349-f006:**
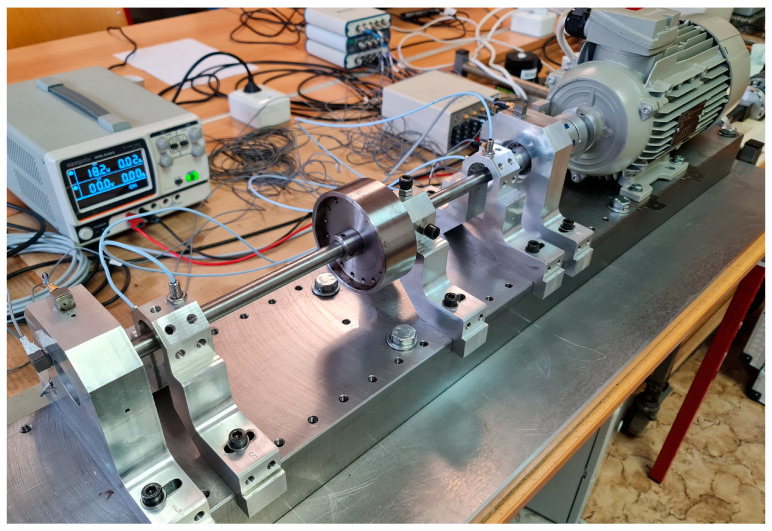
Rotor with the 05un configuration (Disk mounted in the center of the shaft).

**Figure 7 sensors-24-04349-f007:**
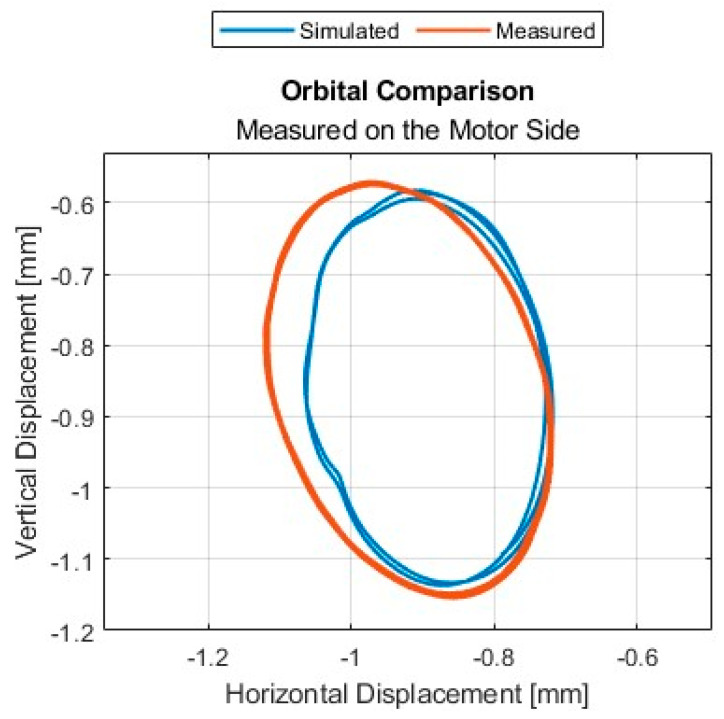
Comparison of numerical and experimental shaft orbital.

**Figure 8 sensors-24-04349-f008:**
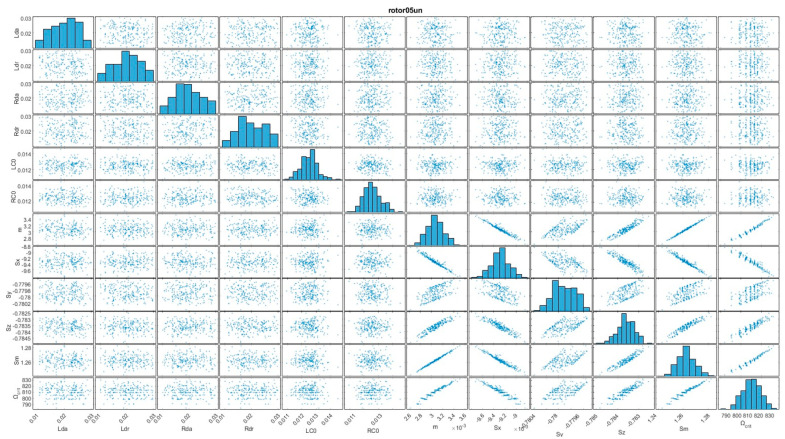
Scatter plot graph of influence between input arguments and analyzed arguments for the rotor05un configuration.

**Figure 9 sensors-24-04349-f009:**
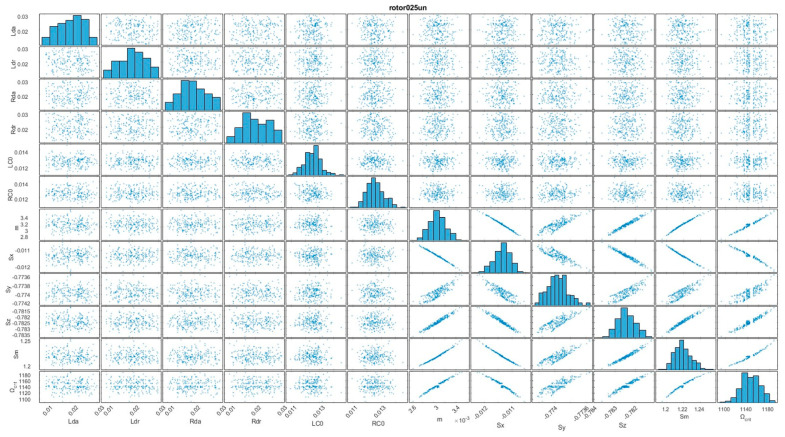
Scatter plot graph of influence between input arguments and analyzed arguments for the rotor025un configuration.

**Figure 10 sensors-24-04349-f010:**
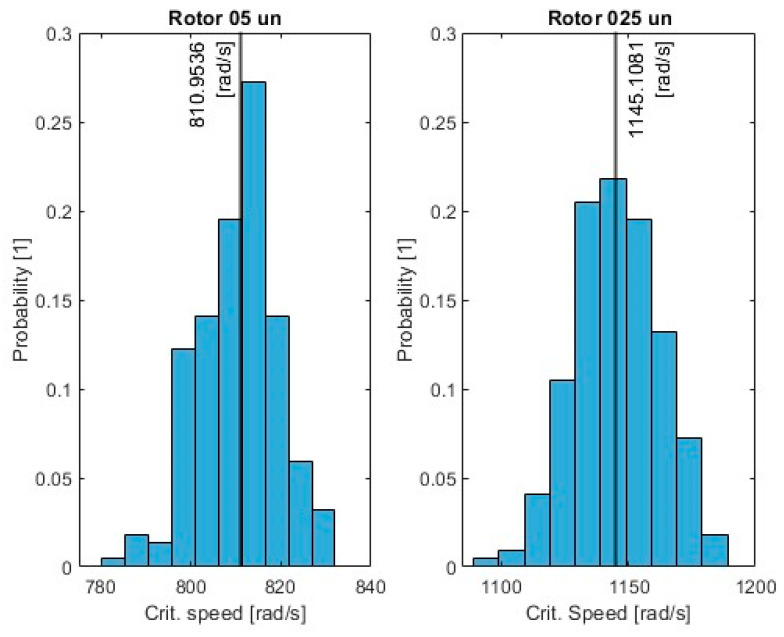
Histograms of the probability distribution of critical speed.

**Figure 11 sensors-24-04349-f011:**
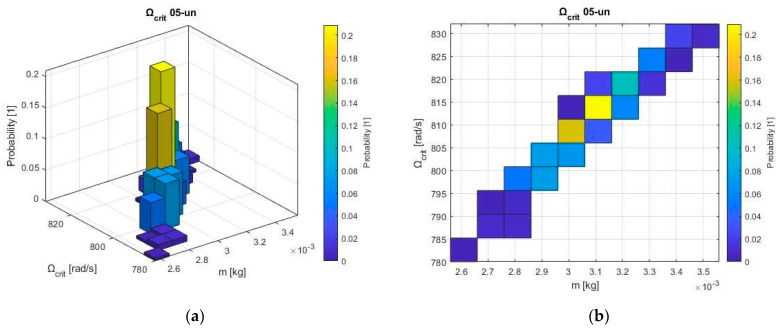
Relation between unbalanced mass and critical speed with the probability distribution of rotor 05un configuration.: (**a**) Bivariate 3D histogram plot; (**b**) Bivariate 2D histogram plot.

**Figure 12 sensors-24-04349-f012:**
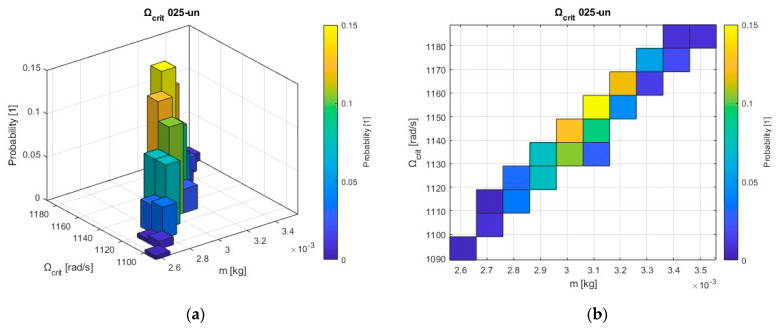
Relation between unbalanced mass and critical speed with the probability distribution for the rotor 025un configuration: (**a**) Bivariate 3D histogram plot; (**b**) Bivariate 2D histogram plot.

**Figure 13 sensors-24-04349-f013:**
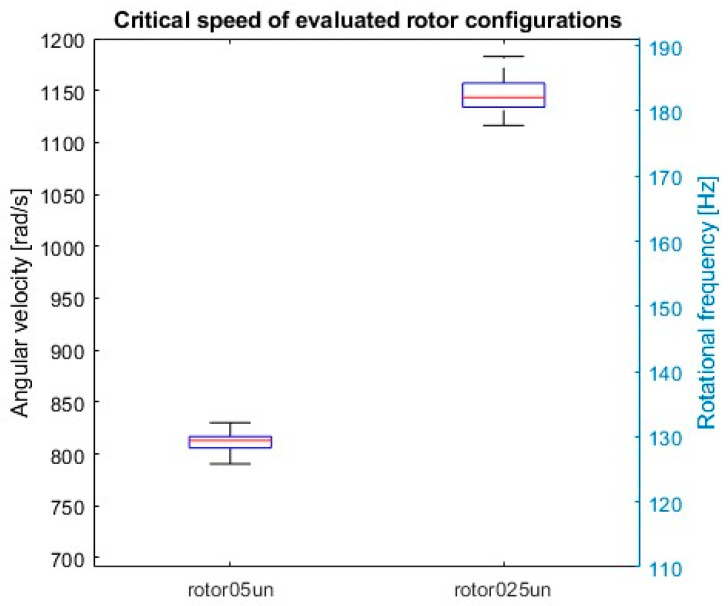
Boxplot of critical speed for evaluated rotor configurations.

**Table 1 sensors-24-04349-t001:** Table of input variables for Monte Carlo analysis.

	Left Bearing			Right Bearing			Disc Mass
Design Variable	Axial Damping Factor [-]	Radial Damping Factor [-]	Clearance [μm]	Axial Damping Factor [-]	Radial Damping Factor [-]	Clearance [μm]	Unbalanced Mass [g]
Range	0.01–0.03	0.01–0.03	5–20	0.01–0.03	0.01–0.03	5–20	2–4

**Table 2 sensors-24-04349-t002:** Quantiles of Critical Speeds (rad/s) and Rotational Frequencies (Hz).

Rotor Configuration	Quantile	f_crit (Hz)	Ω_crit (rad/s)
rotor025un	2.50%	176.76	1110.62
	25%	180.47	1133.93
	50%	181.86	1142.66
	75%	184.16	1157.11
	97.50%	187.02	1175.08
rotor05un	2.50%	125.87	790.89
	25%	128.11	804.93
	50%	129.3	812.44
	75%	129.89	816.15
	97.50%	131.69	827.43

## Data Availability

The data that support the findings of this study are available from the corresponding author upon request.

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
