# Peer review of "Probabilistic Analysis of Critical Speed Values of a Rotating Machine as a Function of the Change of Dynamic Parameters"

_sensors, 2024, doi:10.3390/s24134349_

Round 1
Reviewer 1 Report
Comments and Suggestions for Authors
In this work is performed a Monte Carlo simulation approach using MSC Adams View and Adams Insight to investigate the impact of these uncertainties on the performance of a Laval/Jeffcott rotor model.
The proposal is interesting but some issues have to be addressed.
1. The novelty and contributions are not clear, pleas try to emphasize the most important benefits of this proposal and how the findings lead to improve the operation of rotating system.
2. There are too many small paragraphs in the introduction section, I suggest to group the discusses references and discuss then according to a specific topic, i.e., common techniques used to analyse unbalance, common element analysed under the presence of imbalance, among others.
3. Also, in introduction section, I suggest to include a paragraph to mention those critical effects that are produced due the occurrence of unbalances in rotating machines, this malfunction conditions represent one of the most common problems in industrial scenarios, the following references may be considered:
https://doi.org/10.1007/s10845-019-01483-y ;
https://doi.org/10.1016/j.measurement.2021.109404 ;
https://doi.org/10.1007/s11071-021-06263-z
4. It could be interesting to include a general flow chart that describes the proposed method, this is with the aim of understanding all the performed stages.
5. The whole manuscript is composed by a lot of small paragraphs, please try to write or fusion paragraph to increase the length of paragraphs but maintain the main ideas.
6. Is the proposed method compared with other previous reported methods?
7. What are the limitations of this proposal? Include s brief discussion of them in the conclusion section.
Author Response
Thank you for the review! All changes in article are marked with the red letters (red text).
- The novelty and contributions are not clear, pleas try to emphasize the most important benefits of this proposal and how the findings lead to improve the operation of rotating system.
Mentioned in the introduction last paragraph marked with red color.
2. There are too many small paragraphs in the introduction section, I suggest to group the discusses references and discuss then according to a specific topic, i.e., common techniques used to analyse unbalance, common element analysed under the presence of imbalance, among others.
Small paragraphs discussing the same article joined together so it form a single unit.
3. Also, in introduction section, I suggest to include a paragraph to mention those critical effects that are produced due the occurrence of unbalances in rotating machines, this malfunction conditions represent one of the most common problems in industrial scenarios, the following references may be considered:
https://doi.org/10.1007/s10845-019-01483-y ;
https://doi.org/10.1016/j.measurement.2021.109404 ;
https://doi.org/10.1007/s11071-021-06263-z
References added to the introduction section.
4. It could be interesting to include a general flow chart that describes the proposed method, this is with the aim of understanding all the performed stages.
In the paragraph 3.1. Methodology is added a graphical representation of the workflow.
5. The whole manuscript is composed by a lot of small paragraphs, please try to write or fusion paragraph to increase the length of paragraphs but maintain the main ideas.
Small paragraphs on the same topic joined together and rephrased so the main idea is preserved.
6. Is the proposed method compared with other previous reported methods?
Comparison with a method incorporating Machine learning added to introduction.
7. What are the limitations of this proposal? Include s brief discussion of them in the conclusion section.
Discussion about limitation of proposed method is added to the conclusion section.

Reviewer 2 Report
Comments and Suggestions for Authors
The present work provides a procedure for estimating the critical velocity of a flexible rotor with rigid disk in terms of a probability density function. The combination of scatterplot matrices and histograms facilitated understanding of the behavior and the performance of the rotor configuration.
The paper can be published only if the authors answer to the following remarks:
1. The titles of different papers should not be mentioned in the text of the Introduction, instead the authors of the cited paper and reference number should be mentioned, for example: “Fu et al. [4] presented a novel approach…”
2. At the end of the Introduction the authors must specify the aim and some details about this work and aspects of novelty. Motivation is not sufficiently stated in the introduction part. It should be clarified why they consider this problem and what the advantages of the proposed technique are.
3. The Eqs. (2) and (5) seems to be wrong. The notations xd, yd, θxd are unclear
4. The gyroscopic matrix is not defined. Into Eq.(8) appears Ci but after Eq.(8) appears C.
5. The line between Eqs. (9) and (10) is unclear
6. The Eq.(11) is unclear. All the terms involved in this equation must be explained
7. All the typos from the text of the paper must be removed
Comments on the Quality of English LanguageModerate editing of English language required
Author Response
Thank you for the review! All changes in article are marked with the red letters (red text).
- The titles of different papers should not be mentioned in the text of the Introduction, instead the authors of the cited paper and reference number should be mentioned, for example: “Fu et al. [4] presented a novel approach…”
References changed and articles are now referred to by number.
- At the end of the Introduction the authors must specify the aim and some details about this work and aspects of novelty. Motivation is not sufficiently stated in the introduction part. It should be clarified why they consider this problem and what the advantages of the proposed technique are.
Motivation and advantages of the proposed method are added to the introduction section.
- The Eqs. (2) and (5) seems to be wrong. The notations xd, yd, θxdare unclear
Eq.2. and 5 corrected also with notation.
- The gyroscopic matrix is not defined. Into Eq.(8) appears Cibut after Eq.(8) appears C.
Gyroscopic matrix provided (5) with explanation of terms it involves. Eq 8 (now equation 9 as the gyroscopic matrix was added) corrected.
- The line between Eqs. (9) and (10) is unclear
The line has been changesd so it makes more sense.
- The Eq.(11) is unclear. All the terms involved in this equation must be explained
Terms of the equation explained.
- All the typos from the text of the paper must be removed
The article has been checked, identified typos corrected and sentences written twice, deleted.

Reviewer 3 Report
Comments and Suggestions for Authors
The reviewed article discusses the topic of "Probabilistic analysis of critical speed values of a rotating machine as a function of the change of dynamic parameters". In the opinion of the reviewer, the article is interesting, although::
1) The presented article refers to broadly understood measurement uncertainty. From the presented results, it can be seen that the main unit used in the work is [um]. At the same time, the scope of changes in a given parameter is small. In such a case, the measurement system will have a significant impact on the results. Did the authors analyze the impact of the accuracy of measuring devices on the results?
2) Do environmental conditions and temperatures affect the results?
3) Comparison with other methods would be good.
Author Response
Thank you for the review! All changes in article are marked with the red letters (red text).
1) The presented article refers to broadly understood measurement uncertainty. From the presented results, it can be seen that the main unit used in the work is [um]. At the same time, the scope of changes in a given parameter is small. In such a case, the measurement system will have a significant impact on the results. Did the authors analyze the impact of the accuracy of measuring devices on the results?
No, the influence of the accuracy of measuring devices on the results has not been taken in account. The information about this is added to the paragraph below table 1.
2) Do environmental conditions and temperatures affect the results?
Enviromental conditions as for example the ambient temperature is, were not assumed. This information was added in the paragraph right before the Table 1-
3) Comparison with other methods would be good.
Comparison with a method incorporating Machine learning added to introduction.